# Combined Intervention of Physical Activity, Aerobic Exercise, and Cognitive Exercise Intervention to Prevent Cognitive Decline for Patients with Mild Cognitive Impairment: A Randomized Controlled Clinical Study

**DOI:** 10.3390/jcm8070940

**Published:** 2019-06-28

**Authors:** Hyuntae Park, Jong Hwan Park, Hae Ri Na, Shimada Hiroyuki, Gwon Min Kim, Min Ki Jung, Woo Kyung Kim, Kyung Won Park

**Affiliations:** 1Department of Health Care Science, Dong-A University, Busan 49315, Korea; 2Center for Gerontology and Social Science, National Center for Geriatrics and Gerontology, Obu 474-8511, Japan; 3Health Convergence Medicine Research Group, Biomedical Research Institute, Pusan National University Hospital, Busan 49241, Korea; 4Department of Neurology, Seongnam Center of Senior Health and Bobath Memorial Hospital, Gyeonggi-do 13552, Korea; 5Busan Metropolitan Dementia Center, Busan 49201, Korea; 6Department of Neurology, College of Medicine, Dong-A University, Busan 49201, Korea

**Keywords:** dual-task, randomized controlled trial, dementia, older adults

## Abstract

This study aimed to investigate the association between a dual-task intervention program and cognitive and physical functions. In a randomized controlled trial, we enrolled 49 individuals with MCI. The MCI diagnosis was based on medical evaluations through a clinical interview conducted by a dementia specialist. Cognitive assessments were performed by neuropsychologists according to standardized methods, including the MMSE and modified Alzheimer’s disease Assessment Scale-Cognitive Subscale (ADAS-Cog), both at baseline and at 3 months follow-up. The program comprised physical activity and behavior modification, aerobic exercise, and a cognitive and exercise combined intervention program. Analysis of the subjects for group-time interactions revealed that the exercise group exhibited a significantly improved ADAS-Cog, working memory, and executive function. Total physical activity levels were associated with improvements in working memory function and the modified ADAS-Cog score, and the associations were stronger for daily moderate intensity activity than for daily step count. The 24-week combined intervention improved cognitive function and physical function in patients with MCI relative to controls. Encouraging participants to perform an additional 10 min of moderate physical activity under supervision, during ongoing intervention, may be more beneficial to prevent cognitive decline and improve exercise adherence.

## 1. Introduction

Mild cognitive impairment (MCI) is a state of declined cognitive function and developing Alzheimer’s disease (AD), or it may be present in the preclinical stage of other types of dementia. In this state, the ability to perform daily functional activities is retained, indicating that the patient is at an intermediate stage between normal aging and dementia [1,2]. Therefore, early intervention at the MCI stage can allow patients to retain and improve their cognitive function [3].

Previous randomized controlled trials and meta-analyses have indicated that exercise is associated with improvements in executive function, attention, and processing speed in older adults [4,5]. In addition, aerobic exercise has been shown to enhance brain volume [4,6] and improve functional connectivity between regions of the posterior, frontal, and temporal lobes in healthy older adults [7].

However, a previous meta-analysis reported that a single-bout type of exercise did not improve cognitive function in patients with MCI. Holtzer et al. [8] reported that dual-task exercise (e.g., walking while talking), rather than single-bout exercise, is effective in improving brain function. Previous studies [9] have been limited to only focusing on the function of memory, and the effects of dual-task exercise on different domains of cognitive function, including attention, executive function, and processing speed, remain unclear. Finally, most previous studies have been conducted in a community-based setting, which may be limited by the lack of professionals to perform cognitive and physical function assessments as in a clinical setting.

The present study aimed to investigate the association between a dual-task intervention program and cognitive and physical functions. A single-blinded, randomized, controlled clinical trial was performed to examine the effect of exercise and cognitive combined intervention on the risk factors for dementia in patients with MCI. It was hypothesized that dual-task intervention would have beneficial associations in patients with MCI.

## 2. Materials and Methods

### 2.1. Participants

We recruited 126 patients with MCI who attended a memory and dementia clinic at a university hospital because of memory disturbance or a decline in cognitive function. Seventy-seven were excluded for not meeting the inclusion criteria and 49 agreed to join the intervention program. The MCI diagnosis [10] was based on medical evaluations through a clinical interview by a dementia specialist. The diagnosis was also based on neurological examinations, blood test, brain computed tomography or magnetic resonance imaging, and detailed neuropsychological assessments. In this study, seventy-four participants were excluded from the data analysis, although their feedback was still evaluated because they had a history of stroke or were undergoing treatment for stroke or epilepsy; they were suspected of having other degenerative diseases or mental illnesses based on their medical history; they were depressed or abusing drugs; or they had head injuries, thyroid malfunction, or other medical abnormalities that could impair cognitive function. Non-amnestic MCI was also excluded from the subtypes of MCI classified by the Petersen criteria. A total forty-nine patients with amnestic MCI were enrolled [1]. The patients in a healthy group were >60 years old, could provide information regarding their daily function, and were cognitively normal. After obtaining informed consent from the patients and/or family care providers of the participants, patients were randomly divided into control (*n* = 24) or exercise (*n* = 25) groups (Figure 1). However, two patients dropped-out of each group. The present study was performed in accordance with the International Harmonization Conference guidelines on Good Clinical Practice and was approved by the Dong-A University Hospital Institutional Review Board (IRB No. 14-214) and the IRBs of each center prior to commencement of the study. Prior to participation in the study, all participants or their legally authorized representatives provided written informed consent to participate in the study. This study was registered in the University Hospital Medical Information Network (UMIN) Clinical Trials Registry (No. UMIN000018933).

### 2.2. Neuropsychological Assessment 

General cognitive assessments were performed by neuropsychologists according to standardized methods, including the Korean Mini-Mental State Examination (KMMSE) [11] and modified the Alzheimer’s Disease (AD) Assessment Scale-Cognitive Subscale (ADAS-Cog, range 0–89), both before the intervention and at the 3 months follow-up [12,13]. Symptoms of depression were assessed using the 15 item Korean Version of the Geriatric Depression Scale (SGDS-K) [14,15]. Verbal function was assessed using word fluency tests (category and letter fluency tests) [16]. Detailed neurocognitive function was evaluated using the Seoul Neuropsychological Screening Battery-Dementia version [17,18], which includes assessments of attention, frontal lobe executive function, and verbal functions. Attention was assessed with the digit span test (forward and backward) [19]. Frontal lobe executive function was assessed with the Korean color-word Stroop test [17,18]. 

### 2.3. Physical Function Assessment

Physical and anthropometric variables were measured at before the intervention and after 12 weeks in both groups. Body weight and height were measured to the nearest 0.1 kg and 0.1 cm, respectively, using a body composition analyzer (N20, AIIA Communication, Inc., Korea). Body mass index (BMI) was calculated as weight (kg) divided by height squared (m^2^). 

Arterial blood pressure was measured using a mercury sphygmomanometer after the participants were seated at rest for 10 min. Two measurements were taken at each time point, and the mean was calculated and used for analysis. 

Grip strength was measured using an isometric dynamometer (TKK-5401, Tokyo, Japan) for the dominant arm. The time up and go was measured based on the time required to stand from a standard armchair, walk 2.44 m, turn, walk back 2.44 m, and sit down again at the fastest speed. Walking speed was measured at 5 m walking time, except for 1.5 m each in acceleration and deceleration zones, respectively. To assess balance and sit-to-stand (short physical performance battery), existing test guidelines for each procedure were followed.

### 2.4. Dual-Task Exercise Program

The combined activity program involved 24 weekly 110 min sessions focused on physical and cognitive activities. Between five and six individuals participated in each session conducted by two geriatric exercise specialists and one nurse or occupational therapist. The program combined the physical activity promotion and behavior modification, aerobic exercise, and cognitive and exercise dual-task training program. In the dual-task training, participants conducted cognitive tasks such as word games, performed fast simple numerical calculations, and played a simple memory span game while performing aerobic exercises, what we called *“cognicise”* [20]. Each session featured, in order, a warm-up for 10 min, stretching training for 10 min, aerobic exercise for 20 min, balance exercise for 10 min, aerobic and cognitive dual-task training for 30 min, rest with feedback and promoting daily physical activity education for 20 min, and cooling down for 10 min. The aerobic exercises included stair stepping, endurance walking and stair climbing, and walking on an agility ladder. The target heart rate zone for aerobic exercise during the intervention was at 55%–80% of maximum heart rate (HR), consistent with previous studies. The HR was self-assessed immediately following aerobic exercise based on pulse rate using the Polar S410s heart rate monitor (Polar Electro Oy, Kempele, Finland). The maximum HR was estimated as follows: 207 − 0.7 × age. 

To promote physical activity and healthy behaviors, an exercise specialist provided feedback and encouraged the participants as to how exercise affects dementia, monitored daily physical activities, and advised on how to improve their daily activity with self-directed activity goals. 

### 2.5. Physical Activity Measurements 

Details of the procedure have been described previously [20,21,22,23,24,25]. In brief, an accelerometer (modified Kenz Lifecorder, Suzuken Co., Ltd., Nagoya, Aichi, Japan) was attached to a waist belt on either the left or right side of the body. This device measured the number of steps counts and the intensity of physical activity every 4 s throughout each day for 24 weeks. The intensities were categorized into 11 levels, based on the pattern of the accelerometer signal; the output was expressed in metabolic equivalents (METs) [21]. This recording instrument received detailed appraisals from several groups of investigators [21,22,23]; step counts were determined with an intra-model reliability of 0.998 and an accuracy within ±3% of the actual number of steps taken. Following inspection for inappropriate data recording, the total step counts and intensity categories were calculated over each 24 h period between midnight and the following midnight. The parameters calculated were the average daily period of exercise at an intensity judged as higher than moderate for a typical elderly person (>3 METs). On the basis of individual interviews, we confirmed that the physical activity patterns of our subjects had relatively remained. 

### 2.6. Statistical Analysis

The Shapiro–Wilk test was used to determine the normality of data distribution. The student’s *t*-test was used to assess differences in the baseline (beginning of the intervention) variables. We used the intention-to-treat approach, and between-group comparisons of continuous variables were conducted using linear mixed models after adjusted for the premeasurement covariates for primary and secondary endpoint outcome. Time was treated as a categorical variable, and the models included group, time, and group-by-time interaction as fixed effects, and participants as random effect. For each group, data were expressed as change from baseline (admission) to discharge, determined by the time coefficients (95% CI) of the model. The primary conclusions about effectiveness of the combined intervention were based on between-group comparisons of change in global and prefrontal cognitive function (working memory, processing, and executive function) from baseline to 24 weeks after, as assessed with the ADAS-Cog, KMMSE, digit span test (DST), trail making test-A (TMT), and digit symbol substitution test (DSST) determined by the time-by-group interaction coefficients of the model. All comparisons were two-sided, with an alpha level of 0.05, where the Bonferroni–Holm multiple test adjustment was applied. All statistical analyses were made with SPSS, version 20 (IBM Corp) and R, version 3.2.2 (R Foundation) software. All data were analyzed using the IBM SPSS Statistics version 25.0 software package for Windows (SPSS Inc., Chicago, IL, USA) and SAS statistical software (SAS Institute, 1996).

## 3. Results

### 3.1. Primary Outcomes (Global Cognitive Function)

Analysis of subjects for group-time interactions revealed that the exercise group exhibited a significantly improved modified ADAS-cog score (range 0–89), working memory (DST), and executive function (DSST) scores (Figure 2 and Table 1). The KMMSE score was slightly improved in the intervention group, but not in the control group; however, this did not reach statistical significance, and there was no difference in the processing speed among the two groups. 

### 3.2. Secondary Outcomes 

At baseline, there were no significant differences in physical characteristics among the two groups (Table 2). The within-group analyses showed that body mass did not differ significantly in either group after 12 weeks, nor did BMI. The within-group analyses showed that the waist–hip ratio did not differ significantly from the baseline values in either the exercise or control groups after 12 weeks. However, the within-group analyses showed that appendicular skeletal muscle index (ASMI) was significantly higher than at baseline in the exercise group after 12 weeks.

There were significant improvements in SGDS-K values and habitual physical activity (moderate to vigorous physical activity (MVPA) and step counts compared with scores in the control group at 24 weeks. Compared with the control group, lower-limb physical function such as gait speed, timed up and go test, and sit to stand time showed significant improvement after the intervention. Total physical activity levels were associated with improvements in working memory function and the modified ADAS-cog score, and the associations were stronger for daily duration of activity of moderate intensity than for daily step count. The MVPA during the study duration (i.e., 12 weeks) was 10.1 ± 6.5 min/week in the control group and 22.3 ± 9.3 min/week in the exercise group. Compared with the control group, MVPA durations were longer in the intervention group, even after adjusting for covariates. In the intervention period, the daily accelerometer-assessed time in MVPA increased exponentially to the threshold required for improving function ~22 min/day (150 min/week), with the control group values being slightly decreased (Figure 3). Adherence to the combined program was 95.3 ± 2.3 in the exercise group. Heart rate (HR) during the exercise program was 128.1 ± 5.8 beats/min. This corresponded to an exercise intensity of 70% ± 4% of the age-predicted maximum HR.

## 4. Discussion

The present study investigated the effects of a 24 week multicomponent intervention on the cognitive function and physical function of individuals diagnosed with MCI. In keeping with our initial hypothesis, following adjustments for physical and/or lifestyle-related factors, the multicomponent intervention in patients with MCI had a significant positive effect on both cognitive and physical function. Additionally, time interaction had a significant effect on the primary outcome (global cognition), main secondary outcome (executive function, immediate memory), and other secondary outcomes (gait speed, 5 chair standing time, daily MVPA), but no significant effect on processing speed or grip strength. There were positive effects on global cognitive function, immediate memory and executive function, and lower limb physical function.

The above results, in conjunction with our previously published findings in community-dwelling subjects, may indicate that exercise intervention has positive effects on physical and cognitive function in individuals with MCI [20,26].

A previous systematic review suggested that the evidence of physical exercise on domain-specific cognitive function and psychological outcomes remains unclear, that additional trials with rigorous study design are required to provide further evidence [27], and that the limited evidence for the prevention of cognitive decline or dementia was insufficient [28]. However, another review found that cognitive training had moderate to large benefits in terms of memory-related outcomes [29]. A recent large cohort study reported that a physical activity promotion program, compared with a health education program, did not result in improvements in global or domain-specific cognitive function in sedentary older adults [30]. The results from this study indicate that there appears to be merit in a multicomponent composite approach involving physical activity modification, aerobic exercise, and cognitive exercises for improving cognitive and physical function in older patients with MCI, possibly offering more benefit than a single-bout intervention approach. 

Increasing evidence indicates that acute aerobic exercise, defined as a single bout of exercise, is associated with improved cognitive function, particularly prefrontal cortex-dependent cognition. However, the effects of a single session of exercise on cognitive functioning are generally small. Therefore, besides the type, frequency, and duration over time, even the intensity is a parameter to be significantly considered when evaluating intervention effects. 

The multidomain supervised intervention used in the present study was feasible and safe. The dropout rates were low and adherence to the intervention domains was high. To the best of our knowledge, this is the first randomized control trial to investigate the impact of a combination of physical activity, aerobic exercise, and cognitive training (not only on cognitive function but also on physical function) and objectively monitored physical activity during the intervention period for motivating and dose adjustment of exercise in patients with MCI. The training was relatively intensive and performed 2 days per week, under supervision. Epidemiological studies have suggested the possibility that unsupervised moderate-intensity aerobic activity, consistent with American College of Sports Medicine recommendations, or subject-choice physical or cognitive activities, may also be beneficial in reducing age-related cognitive decline. However, evidence was not available when the study investigating the direct effects of a combined physical and cognitive training in patients with MCI commenced. Therefore, it was considered important to provide supervised, reproducible activities. In addition to this, the presence of daily activity supervision and feedback contributed to high compliance in the training performed by subjects. The fact that training activities were inspired by habitual activities made it possible for each participant to identify matching daily activities.

Numerous longitudinal studies have reported a reduced risk of cognitive decline in physically active older adults [31,32] and protected cognition in patients with MCI with higher levels of physical activity [33,34]. A previous study [33] suggested beneficial effects on global cognition and cognitive domains, which are highly relevant for everyday activities. Similarly, regular physical activity can reduce this risk and provide other physical and possibly mental health benefits. However, the dose–response relationship between physical activity and cognition is yet to be fully elucidated and the majority of adults are not active at the recommended levels. In our intervention, it appeared that professional advice and supervised guidance with continued support can encourage individuals to be more physically active in daily life. A notable finding of our study was that the habitual moderate to vigorous physical activity increased by approximately 10 min per day in the intervention group, whereas it remained constant or decreased slightly in the control group during the course of the study. This increase in daily activities in the intervention group resulted in increased physical and cognitive function. Therefore, intervention studies should obtain information on the activity levels of an individual, activity began, intervention duration, and baseline activity. Our combined intervention program enhanced daily activity levels in patients with MCI, and the results suggest that these protective effects and the exercise intervention promoting an additional 10 min daily MVPA, may be beneficial to prevent cognitive and physical decline. Of note, the activity of the participants improved sufficiently to meet the minimum physical activity guidelines [35,36]. Therefore, it is possible that these additional activities may have affected the findings.

In the pre-dementia stage, converted patients showed lower global cognitive function and greater semantic memory impairment (verbal fluency and vocabulary) than non-converted patients [37]. From a clinical perspective, working memory assessment is important. For example, working memory performance has predictive value in patients with MCI with respect to the development of dementia [38,39]. Although episodic deficits during disease progression have been widely investigated and are the benchmark of a probable diagnosis of AD, more recent research has investigated working memory and executive function decline during MCI, also referred to as the preclinical stage of AD. Monitoring the performance on working memory and executive function tasks to track cognitive function may signal progression from normal cognition to MCI to AD. A previous study reported the progression of MCI toward AD in the intervention group consisted of disintegration of the compensatory networks observed by under activation of the lateral PFC, precuneus and posterior parietal regions, which are all involved in executive function, compared with non-impaired age-matched controls [40].

In the present study, the physical and cognitive combined intervention program improved physical fitness, physical activity, executive function test scores, and working memory scores in older patients with MCI than in the control group. This intervention enhanced prefrontal cognitive function gains in patients with MCI. Brain volume shrinks 0.5%–1% annually after the age of 65 years, and hippocampal volume shrinks by 1%–2% annually in older adults without dementia, increasing the risk of cognitive impairment [41]. Despite this mechanism, the underlying beneficial effects of exercise to reduce the risk of AD remains unclear. A recent study [42] found that exercise boosted hippocampal neurogenesis, reduced amyloid-β levels, increased the levels of brain-derived neurotrophic factor, and improved memory in an AD mouse model. The ablation of neurogenesis prevented the beneficial effects of exercise in this model. In the human study, a clinical trial of exercise training was effective at reversing hippocampal volume loss, cognitive decline, and cardiovascular physical performance in late adulthood with moderate-intensity exercise [43]. 

Another possible explanation for the impact of exercise on cognition is that AD is marked by changes in cerebral blood flow (CBF); patients with AD show a 40% decrease in global blood flow compared with healthy controls [44]. The decrease in CBF occurs in individuals with MCI prior to its transition to AD [45]. In healthy individuals, cardiorespiratory fitness mediates the age-related gray matter CBF [46]. Therefore, increasing physical fitness by aerobic exercise assists in the prevention or slowing of pathological cognitive decline by an increase in CBF. 

The strengths of this randomized controlled trial include a clinical research design designed to clarify the possibility to maximize cognitive benefits using the components of cognitive training combined with the promotion of physical activity. The findings of this trial can be generalized or applied to hospital clinics or regional dementia centers with trained instructors. In the previous study, the long-term exercise and cognitive-composite program improved the cognitive function for 1 year or 6 month. However, in the present study, the 6 month well-supervised exercise intervention increased cognitive function and physical function. 

The present study had several limitations. The sample size was small and replication with a larger group of adults with MCI would be beneficial. Other limitations include unknown group differences in the risk factors for cognitive decline and AD, including apolipoprotein E ε4 genotypes [47] and inflammation [48], although there were no significant differences between the groups in hypertension, diabetes mellitus, medications, physical performance or depressive moods. In addition, it is possible that the improvement in the exercise group resulted from the social contact received by the intervention group. This possibility is unavoidable and should be addressed in future studies.

## 5. Conclusions

As more is learned regarding the underlying mechanisms and the nuanced effects exposed by analyzing larger samples, the designing and tailoring of interventions can be more effective. The 24 week combined intervention improved cognitive function and physical function in patients with MCI relative to controls. Encouraging participants to perform an additional 10 min of moderate physical activity under supervision, during ongoing intervention, may be more beneficial to prevent cognitive decline and improve exercise adherence.

## Figures and Tables

**Figure 1 jcm-08-00940-f001:**
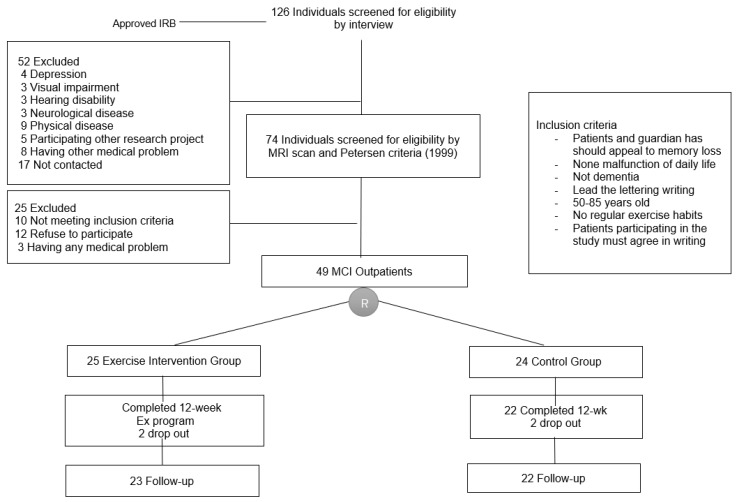
Subject flow diagram from initial contact through to study completion.

**Figure 2 jcm-08-00940-f002:**
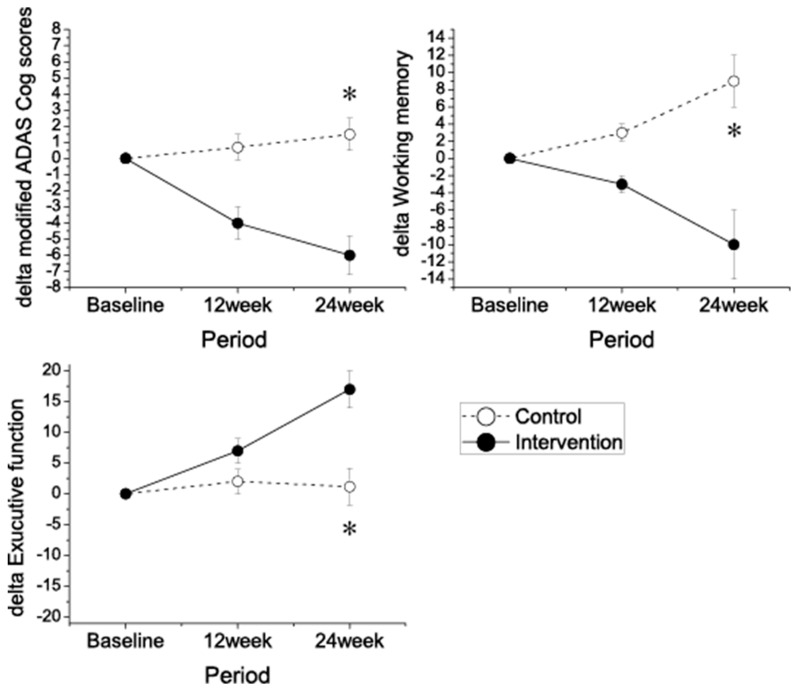
Change in cognitive performance during the 24 week intervention. Mean change in modified the Alzheimer’s Disease Assessment Scale-Cognitive Subscale ADAS-Cog score from baseline (negative differences correspond to lower scores, indicating performance improvement). Error bars are SEM; *p*-value (*p* < 0.05) from mixed-model repeated measure analysis, group × time interaction. * *p* < 0.05.

**Figure 3 jcm-08-00940-f003:**
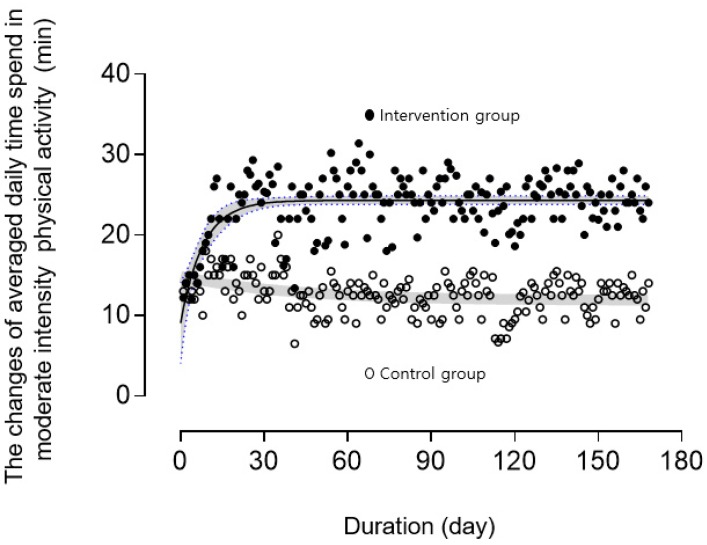
Changes in averaged daily moderate to vigorous physical activity during the intervention period. Black and gray lines show curve-fitting in both linear and exponential fit and 95% confidence band. Both groups wore waist-mounted accelerometers during the intervention period.

**Table 1 jcm-08-00940-t001:** Adjusted comparisons of change in measurements from baseline and 24 in intervention group and control group.

Variables	Intervention Group		Control Group		Between-Group Difference (95% CI)	*p* Value between Groups
	(Before)	(Follow-up)	(Before)	(Follow-up)		
***Primary outcome***						
Modified ADAS-Cog	26.2 ± 2.9	24.6 ± 3.3	25.7 ± 3.1	26.1 ± 2.7	−1.9 (−0.9 to −2.3)	<0.01
MMSE	24.6 ± 2.6	24.8 ± 3.7	24.4 ± 3.1	24.2 ± 3.0	0.3 (−0.2 to 0.9)	0.06
***Secondary outcome***						
Working memory (Digit Span Scores)	2.7 ± 0.2	2.4 ± 0.2	2.6 ± 0.3	2.9 ± 0.2	−0.6 (−0.3 to 1.6)	0.02
Processing speed (TMT-A)	25.3 ± 7.1	23.1 ± 6.3	24.7 ± 6.2	24.1 ± 6.7	−0.4 (0.2 to −1.0)	0.1
Exucutive function (SDST)	30.7 ± 8.0	36.3 ± 8.7	34.2 ± 10	34.7 ± 8.1	−6.3 (−4.2 to −8.4)	<0.01
GDS	2.6 ± 1.1	2.5 ± 1.0	2.5 ± 0.9	2.7 ± 1.2	−0.6 (−0.1 to −1.2)	0.02
Gait speed (m/s)	1.06 ± 0.3	1.11 ± 0.4	1.08 ± 0.3	1.05 ± 0.3	0.4 (0.0 to 0.9)	0.02
Grip strength (kg)	27.8 ± 6.2	28.6 ± 5.4	26.7 ± 7.1	29.0 ± 6.8	0.5 (−0.7 to 1.6)	0.132
Timed up and go (s)	10.1 ± 3.1	8.9 ± 3.4	9.7 ± 4.1	9.5 ± 3.9	−0.8 (−0.4 to −1.4)	<0.01
Sit to standing time (s)	18.0 ± 5.2	16.9 ± 4.5	17.3 ± 4.7	17.7 ± 4.8	0.9 (0.4 to 1.7)	<0.01
MVPA (min/day)	12.2 ± 7.7	22.3 ± 9.3	11.9 ± 6.3	10.7 ± 6.5	10.6 (7.3 to 20.1)	<0.01
Step counts (steps/day)	4876 ± 867	7893 ± 1001	4794 ± 763	4210 ± 861	2996 (1307 to 3816)	<0.01

Abbreviations: ADAS-Cog, Alzheimer’s Disease Assessment Scale-Cog; MVPA, moderate and vigorous physical activity; GDS, Geriatric Depression Scale; TMT, train making tese, SDST, Symbol–Digit Substitution Test: MVPA, Moderate to vigorous physical activity, Values are given as means± SD. *P* < 0.5 was considered significant.

**Table 2 jcm-08-00940-t002:** Baseline characteristics of socio-demographic, physical characteristics, daily physical activity, mental and cognitive function.

Variables	Exercise Group (*n* = 25)	Control Group (*n* = 24)	*p* Value
**Socio-demographic**			
Age (years)	70.55 ± 6.46	72.76 ± 5.37	0.239
Male, *n* (years)	8 (40.0%)	7 (33.3%)	-
Education level (years)	7.15 ± 2.94	7.05 ± 3.28	0.971
Job, *n* (%)	2 (10.0%)	3 (14.3%)	0.95
Smoking (yes), *n* (%)	3 (15.0%)	1 (4.8%)	0.281
Alcohol consumption (yes), *n* (%)	5 (25.0%)	1 (4.8%)	0.077
**Physical Characteristics**			
Height (m)	1.55 ± 0.09	1.56 ± 0.07	0.689
Weight (kg)	55.17 ± 5.46	57.50 ± 6.51	0.224
Body Mass Index (kg/m^2^)	23.05 ± 2.29	23.70 ± 2.07	0.342
Waist hip ratio	0.87 ± 0.05	0.87 ± 0.05	0.329
Fat (%)	29.86 ± 9.99	30.22 ± 9.99	0.886
ASMI (kg/m^2^)	7.24 ± 1.60	7.33 ± 1.27	0.839

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
