# Peer review of "Combined Intervention of Physical Activity, Aerobic Exercise, and Cognitive Exercise Intervention to Prevent Cognitive Decline for Patients with Mild Cognitive Impairment: A Randomized Controlled Clinical Study"

_jcm, 2019, doi:10.3390/jcm8070940_

Reviewer 1 Report

General comments:

1.     The acronym PACE should be reconsidered, in order to avoid confusion. An Australian team published a randomized controlled trial named PACE Study in 2015 (Vidovich et al. 2015, Am J Geriatr Psychiatry 23:360-372). It was a study in patients with MCI and the acronym PACE stood for Promoting Healthy Ageing with Cognitive Exercise.

2.     Reference to tables and figures are missing in the text.

3.     Many abbreviations are used in the text without explanation. When an abbreviation is used first, it should be introduced with the full term. It is not sufficient that some abbreviations can be found in a footnote to a table.

4.     An overall English language check seems advisable.

Specific comments:

5.     Title: The words "for at-risk" do not seem to fit here.

6.     Abstract, line 20: Mentioning here only the number of subjects screened may be misleading. More important is the number of randomized subjects, because the results and the conclusion are derived from the latter.

7.     Materials and Methods, Section 2.1. Participants, line 64: To clearly distinguish here the number of patients screened from those randomized, it would be preferable to say: "We screened 126 patients with MCI …"

8.     Materials and Methods, Section 2.1. Participants, line 64ff: Were any of the published diagnostic criteria for MCI applied?

9.     Materials and Methods, Section 2.1. Participants, line 74: Does [1] mean that the criteria published by Petersen in 1999 were applied or is it a reference to Figure 1 (which seems to be missing in this paragraph)?

10.  Materials and Methods, Section 2.1. Participants, line 81: Common clinical criteria for MCI require that overall cognitive function is still intact and that everyday functioning is largely intact; only instrumental activities of daily living may be slightly impaired. Patients with MCI would therefore usually be able to give informed consent. If patients were included who had a legal representative, this may cast doubt on the diagnosis of MCI. There should be a few words explaining and justifying this.

11.  Materials and Methods, Section 2.2. Neuropsychological assessment, line 86: Which version of the ADAS-cog was used? In Figure 2, along the abscissa of the upper left figure, it reads "delta modified ADAS Cog scores". Did you use the original 11-item form or one of the extended forms that have been used in MCI research?

12.  Results, Primary outcomes, line 163: I do not understand this part of the sentence: "but the statistical significance was not obtained baseline and after the intervention". It is shown in Table 2 that the between-group difference did not quite reach significance. But what exactly does the reference to baseline mean?

13.  Results, Secondary outcomes: Here are some examples of abbreviations not explained before (e.g. ASMI, MVPA).

14.  Discussion, first paragraph, line 192: There seems to be a misspelling: "5 ch standing time".

15.  Discussion, third paragraph, line 203: There seems to be an intrusion. The authors of the LIFE study describe their program as a "physical activity program", not as a physical activity medication program. There was no medication as part of the program.

16.  Discussion, sixth paragraph, line 232: It is true, that the multi-domain program of the FINGER Study also comprised physical activity and cognitive training, but it was a multi-domain program that also comprised diet changes and vascular risk monitoring.

17.  Discussion ,sixth paragraph, lines 242-243: This sentence is difficult to understand.

18.  Discussion, sixth paragraph, lines 247-248: The references Haskell et al., 2007, and O'Donovan et al., 2010, should be added to the list of references and integrated in the numbering system.

19.  Discussion, eighth paragraph, line 263-264: This sentence needs re-wording. It should express that the various cognitive and physical abilities improved more in the intervention group than in the control group.

20.  Figure 1: The upper left box is incomplete. What is visible, accounts for only 35 of 52 excluded patients.

21.  Table 2: Please check the first line of the table. Should not "Exercise group" stand above the two columns headed "Intervention group" and "Control group" above the two columns headed "Control group"?

22.  Figure 3: Please check the second sentence of the legend. The figure does not look like representing "combined intervention and control groups".

Author Response

We would like to thank the referees for their favorable reviews and for the comments which have contributed to improvement of the paper.

General comments:

1.   The acronym PACE should be reconsidered, in order to avoid confusion. An Australian team published a randomized controlled trial named PACE Study in 2015 (Vidovich et al. 2015, Am J Geriatr Psychiatry 23:360-372). It was a study in patients with MCI and the acronym PACE stood for Promoting Healthy Ageing with Cognitive Exercise.

The reviewer’s suggestion is appropriate. We have changed PACE to combined (physical + cognitive) intervention program

2. Reference to tables and figures are missing in the text.

Change made, according to the referee.

3.           Many abbreviations are used in the text without explanation. When an abbreviation is used first, it should be introduced with the full term. It is not sufficient that some abbreviations can be found in a footnote to a table.

Thank you for your appropriate suggestion, and we have changed

4.             An overall English language check seems advisable. Specific comments:

Thank you for your suggestion. We made

5. Title: The words "for at-risk" do not seem to fit here.

    Change made, according to the referee.

6. Abstract, line 20: Mentioning here only the number of subjects screened may be misleading. More important is the number of randomized subjects, because the results and the conclusion are derived from the latter.  (초록에 대상자 기입)

 Thank you for your appropriate suggestion, and we have changed

7. Materials and Methods, Section 2.1. Participants, line 64: To clearly distinguish here the number of patients screened from those randomized, it would be preferable to say: "We screened 126 patients with MCI …"

     Thank you for your very appropriate suggestion, and we have changed with your suggestion

8. Materials and Methods, Section 2.1. Participants, line 64ff: Were any of the published diagnostic criteria for MCI applied?  문헌 이전

9. Materials and Methods, Section 2.1. Participants, line 74: Does [1] mean that the criteria published by Petersen in 1999 were applied or is it a reference to Figure 1 (which seems to be missing in this paragraph)?

Thank you for your appropriate suggestion, and we have changed with your suggestion.  We used MCI criteria based on the clinical criteria our previous study and we added the previous reference (Park et al. Geriatr Gerontol Int 2017, 17(10):1603-1609 ).

10. Materials and Methods, Section 2.1. Participants, line 81: Common clinical criteria for MCI require that overall cognitive function is still intact and that everyday functioning is largely intact; only instrumental activities of daily living may be slightly impaired. Patients with MCI would therefore usually be able to give informed consent. If patients were included who had a legal representative, this may cast doubt on the diagnosis of MCI. There should be a few words explaining and justifying this.

The reviewer’s comments is appropriate. In our study, the informed consent process took about 15 min-40min to the all participants and family caregivers. After fully providing informed consent by specialist, we obtained the informed consent from patient and/or family member.  We added the few words for this

11. Materials and Methods, Section 2.2. Neuropsychological assessment, line 86: Which version of the ADAS-cog was used? In Figure 2, along the abscissa of the upper left figure, it reads "delta modified ADAS Cog scores". Did you use the original 11-item form or one of the extended forms that have been used in MCI research?

We used the modified AD Assessment Scale-cognitive subscale and we revised the sentence and additional reference.

(The primary efficacy outcome was the change from baseline to PI in the modified AD Assessment Scale-cognitive subscale (ADAS-Cog, range 0–89) [25].

reference: Doody RS, Ferris SH, Salloway S, Sun Y, GoldmanR, Watkins WE, Xu Y, Murthy AK: Donepezil treatment of patients with MCI: a 48-week randomized, placebo-controlled trial.

Neurology 2009; 72: 1555–1561

12. Results, Primary outcomes, line 163: I do not understand this part of the sentence: "but the statistical significance

as not obtained baseline and after the intervention". It is shown in Table 2 that the between-group difference did not quite reach significance. But what exactly does the

reference to baseline mean?

Thank you for your valuable suggestion.  We change the “baseline” to before, and change the as follows:

 The Korean version of mini-mentel state exam (KMMSE) score was improved in the intervention group compared to the control group before and after the intervention, however, this did not reach statistical significance

13. Results, Secondary outcomes: Here are some examples of abbreviations not explained before (e.g. ASMI, MVPA). 14. Discussion, first paragraph, line 192: There seems to be a misspelling: "5 ch standing time".

Thank you for your good comments, and we made as your suggestion.

15. Discussion, third paragraph, line 203: There seems to be an intrusion. The authors of the LIFE study describe their program as a "physical activity program", not as a physical activity medication

program. There was no medication as part of the program. 

Thank you for your appropriate comments.  We misspelled the modification, and we change “physical activity promotion program”. 

16. Discussion, sixth paragraph, line 232: It is true, that the multi-domain program of the FINGER Study also comprised physical activity and cognitive training, but it was a multi-domain program that also comprised diet changes and vascular risk monitoring.

Your comment is appropriate, and we change the previous evidence to Lautenschlager NT, Cox KL, Flicker L, Foster JK, van Bockxmeer FM, Xiao J, Greenop KR, Almeida OP: Effect of physical activity on cognitive function in older adults at risk for Alzheimer disease: a randomized trial. JAMA 2008, 300(9):1027-1037.32

17. Discussion ,sixth paragraph, lines 242-243: This sentence is difficult to understand.

We changed the sentence (intervention group undertaken continuously for approximately 10 minutes per day more during the intervention period, whereas the physical activity of the control group was constant or decreased after the intervention compared to the before the intervention.)

18. Discussion, sixth paragraph, lines 247-248: The references Haskell et al., 2007, and O'Donovan et al., 2010, should be added to the list of references and integrated in the numbering system.

We added the references in the list of reference.

19. Discussion, eighth paragraph, line 263-264: This sentence needs rewording. It should express that the various cognitive and physical abilities improved more in the intervention group than in the control group.

Thank your meaningful comments. We reworded the sentences.  

20. Figure 1: The upper left box is incomplete. What is visible, accounts for only 35 of 52 excluded patients.

We changed

21. Table 2: Please check the first line of the table. Should not "Exercise group" stand above the two columns headed "Intervention group" and "Control group" above the two columns headed "Control group"?

Thank you for your good comments. We had mistaken, and we exchanged a changed tables.

22. Figure 3: Please check the second sentence of the legend. The figure does not look like representing "combined intervention and control groups".

We changed

Thank you for your valuable and appropriate comments for our manuscript.

Reviewer 2 Report

The study is well conducted, well presented and reports results on the very important topic of the effects of dual task on cognitive performance. The main limitations are described. so the paper may be published in its present form, provided style and language errors are corrected throughout the text. 

Author Response

Thank you for your kind comments.

Round  2

Reviewer 1 Report

Thanks for revising the manuscript. In a few sentences the modifications led to slight errors in the word order.

Abstract, sentence on cognitive assessments: It should read "… the Korean Mini-Mental State Examination and the modified The Alzheimer's Disease (AD) Assessment Scale-Cognitive Subscale …"

Section 2.2. Neuropsychological assessment, first sentence: It should read "… the Korean Mini-Mental State Examination (KMMSE) [11] and the modified The Alzheimer's Disease (AD) Assessment Scale-Cognitive Subscale (ADAS-Cog, range 0-89), both at before the intervention and at the 3 months follow-up[12, 13]." (Intoducing the abbreviation for KMMSE here would make it unnecessary to use the full term later in the text.)

Section 3. Results, primary outcomes, second sentence: I suggest changing the sentence as follows: "The KMMSE score was slightly improved in the intervention group, but not in the control group; however, this did not reach statistical significance." The slight difference in KMMSE scores at baseline was certainly not statistically significant and probably not clinically relevant.

Section 4. Discussion, middle of the sixth paragraph: The sentence on habitual moderate to vigorous physical activity seems to summarize what is shown in Figure 3. In this case, would rewording as follows express it correctly? "A notable finding of our study was that the habitual moderate to vigorous physical activity increased by approximately 10 minutes per day in the intervention group, whereas it remained constant or decreased slightly in the control group during the course of the study."

Table 2: In the first line, the "l" in Control got lost.

Author Response

Thanks for revising the manuscript. In a few sentences the modifications led to slight errors in the word order.

I cordially appreciate for your revision and modification. 

Abstract, sentence on cognitive assessments: It should read "… the Korean Mini-Mental State Examination and the modified The Alzheimer's Disease (AD) Assessment Scale-Cognitive Subscale …"

Section 2.2. Neuropsychological assessment, first sentence: It should read "… the Korean Mini-Mental State Examination (KMMSE) [11] and themodified The Alzheimer's Disease (AD) Assessment Scale-Cognitive Subscale (ADAS-Cog, range 0-89), both at before the intervention and at the 3 months follow-up[12, 13]." (Intoducing the abbreviation for KMMSE here would make it unnecessary to use the full term later in the text.)

We modified as suggested

Section 3. Results, primary outcomes, second sentence: I suggest changing the sentence as follows: "The KMMSE score was slightly improved in the intervention group, but not in the control group; however, this did not reach statistical significance." The slight difference in KMMSE scores at baseline was certainly not statistically significant and probably not clinically relevant.

Thank you for your appropriate modification. We changed as suggested

Section 4. Discussion, middle of the sixth paragraph: The sentence on habitual moderate to vigorous physical activity seems to summarize what is shown in Figure 3. In this case, would rewording as follows express it correctly? "A notable finding of our study was that the habitual moderate to vigorous physical activity increased by approximately 10 minutes per day in the intervention group, whereas it remained constant or decreased slightly in the control group during the course of the study."

Thank you for your good suggestion. We modified as suggested

Table 2: In the first line, the "l" in Control got lost.

Changed the table as your comment